# Utility and use of accuracy cues in social learning of crowd preferences

**Jaeseob Lim**, **Sang-Hun Lee***

Department of Brain and Cognitive Sciences, Seoul National University, Seoul, South Korea

* visionsl@snu.ac.kr

**Data Availability Statement:** All analysis script and data files are available from the Open Science Framework database (DOI: 10.17605/OSF.IO/BVHKA).

**Funding:** JL and SL were supported by the Brain Research Programs and Basic Research

## Abstract

Despite limited information and knowledge, we personally form beliefs about certain properties of objects encountered in our daily life—popularity of a newly released movie, for example. Since such beliefs are prone to error, we often revise our initial beliefs according to the beliefs of others to improve accuracy. Optimal revision requires modulating the degree of accepting others' beliefs based on various cues for accuracy—number of opinions, for example—such that the more accurate others' beliefs are, the more we accept them. Although previous studies have shown that such accuracy cues can influence the degree of acceptance during social revision, they primarily investigated problems with 'factually correct' answers, and rarely problems with 'socially correct' answers. Here we examined which accuracy cues are objectively useful (utility of cues), and how those cues are used (use of cues), in the social revision of people's beliefs about problems with 'socially correct' answers. We asked people to estimate the 'shared preferences (SPs)' for sociocultural items, the answers to which are determined by socially aggregated beliefs—how popular an abstract painting will be among a large crowd, for example—and then to revise their initial estimates after being exposed to other people's estimates about the same items. We considered 'confidence', 'agreement among estimates', and 'number of estimates' as accuracy cues. We found that, while all three cues validly signaled the accuracy of SP estimates, only the 'number' cue has a significant utility, but the other cues are much less useful for optimal revision. Nevertheless, people used the cues of 'agreement' and their own 'confidence' to the extent comparable to that of the 'number' cue. Our findings suggest that the utility and use of accuracy cues for problems with 'socially correct' answers differ from those with 'factually correct' answers, as follows: (i) confidence does not have a significant utility and (ii) but people use their own confidence while ignoring others' confidence.

## Introduction

People have different tastes in many things in life. Some prefer Chardonnay to Sauvignon blanc, while others have different preferences altogether. Nonetheless, making decisions in social contexts often requires us to gain access to the representative preferences of a crowd of interest, which is called "shared preferences" (SP) [1–3]. For example, we ask ourselves which

Laboratory Program through the National Research Foundation of Korea (NRF) funded by the Ministry of Science and ICT of South Korea (http://english. msip.go.kr/english/main/main.do). Grant numbers are NRF-2015M3C7A1031969, NRF-2017M3C7A1047860 for the Brain Research Programs and NRF-2018R1A4A1025891 for the Basic Research Laboratory Program. The funders had no role in study design, data collection and analysis, decision to publish, or preparation of the manuscript.

**Competing interests:** The authors have declared that no competing interests exist.

wine to bring when visiting a neighbor's home for the first time, which music to play at a work party, or which actor to cast for a movie that we are about to produce. In such situations, accurate estimation of SP can benefit our daily social life and help us to predict consumers' preferences in the market.

Unfortunately, our personal estimation of SP is prone to error; relying on our common-sense knowledge about social standards of valuation (e.g., "People find symmetric faces most attractive" [3]) or projecting our own preferences onto a crowd (e.g., "People living in my neighborhood will like the wine I like" [4,5]) often leads to biased estimation [6]. Furthermore, personally formed estimates of SP are also accompanied with substantive degrees of uncertainty and can thus be subject to revision [7]. For these reasons, we often turn to others to correct our personally formed estimation of SP for inaccuracy, such as asking our friends or the Internet, "Will this item be liked by a crowd?" [8–11]. Here arises the following important question: What is the ideal way to accept SP estimates from others, such that our revised SP estimates come close to the actual SP values? One crucial factor that should be considered in addressing this question is that others' SP estimates encountered in our daily social interactions can vary greatly in their accuracy, much like our own personally formed SP estimates do. For example, imagine that we personally formed a belief that the crowd would like a recently released movie and then had encounters with i) one of our friends, who told us that the crowd seemed to dislike the movie, and ii) three friends, who told us that the crowd seemed to dislike the movie. Given that knowing the actual SP is, by definition, a matter of knowing the 'socially correct' answer [12,13], the aggregated opinion of the three friends is likely to be more accurate than the opinion of the one friend. In other words, the number of estimates can be a cue for estimation accuracy. Then, considering that the optimal degree of accepting others' opinion is determined by the accuracy of that opinion [14–16], the optimal revision strategy will be to modulate the degree of acceptance according to the number of friends by revising our initial SP estimate of the movie ("people will like it") toward the friends' SP estimate ("people won't like it") by a larger degree when being told from three friends than from a single friend. If a certain cue, such as the "number of estimates" in the above example, carries valid information about estimation accuracy, the cue can be referred as an 'accuracy cue'. In that case, optimal revision likely to, not always though, involve modulating the degree of acceptance based on that cue, which means that cue has "utility" in this task. To reveal the ideal method for social revision, it is important to identify cues that have 'utility.' However, this should be distinguished from whether those cues are actually 'used' by people during social revision. It is possible that people do not 'use' a certain cue even though that cue has 'utility', and vice versa. So, in this work we investigate them separately: we first identified accuracy cues that have 'utility' and second, examined whether people actually 'use' them in social revision of opinions for problems with 'socially correct' answers.

Many previous studies have investigated how people use accuracy cues in social revision, particularly in the context of advice-taking in 'matter-of-fact' problems that have factually correct answers (see [17,18] for a review). However, few studies have addressed this issue for problems that have 'socially correct' answers, such as SP estimation; only a few studies have focused on accuracy cues that are related to the characteristics of an advisor, such as 'expertise' [12,13,19]. Given that the utility and use of accuracy cues vary greatly depending on the type of problem [13,19,20], these should be evaluated in the context of problems with 'socially correct' answers.

We focused on three candidate accuracy cues, namely, "how many other people contribute to an estimation", "to what extent other people agreed to each other", and "how confident people are about an estimation," which we will refer to as 'number', 'agreement', and 'confidence' cues hereafter. These three cues were chosen for the following reasons. First, they vary

considerably across situations in which we are required to learn SP values in our daily social life. For example, we can get many other's opinions in some situation, but in other, we can only get advice from only one person. Second, they are important accuracy cues for social revision in problems with 'factually correct' answers [17,18,21–25], but their use in problems with 'socially correct' answers has not yet been examined. Lastly, previous studies have suggested that the utility of these three accuracy cues, particularly agreement and confidence cues, depend on how people's estimates are distributed or the type of the problem (judgmental vs. intellective). For instance, the utility of the agreement cue is known to be highly dependent on the distribution of estimates, whereby increasing the degree of accepting others' estimates as estimates become more similar is not the optimal strategy for estimates that sampled from a normal distribution, but is the optimal strategy for estimates that sampled from distribution with a high degree of kurtosis [24–26]. Likewise, the utility of the confidence cue has been reported to become pronounced for simple or concrete problems with definitive answers, such as memory or intellective problems, and weak or negligible for complex or abstract problems that have ambiguous answers, such as general knowledge or judgmental problems (e.g., "Which of these thrillers from 2006 would be best to watch?") [13,27,28]. Given these previous reports, it is necessary to examine whether agreement and confidence cues have utility in social revision of opinions for problems with 'socially correct' answers.

We adapted a real-world situation of social revision to a laboratory setup. In this experiment, we asked participants to estimate SP values for sociocultural items, twice for each item, before and after being exposed to the SP estimates of a few (1~3) others for the same items like in previous advice-taking studies [9,11,25,29]. While performing the task, participants also reported how confident they felt about their estimates. And participants were exposed to randomly picked other's actual estimates and their confidence, so that the number of others' estimates and the agreement between those estimates varied across trials. Then, we examined whether the number, agreement, and confidence cues provide information about SP estimation accuracy across trials and are useful for SP revision, and evaluated how people use these cues to improve the accuracy of SP estimation through social revision. We found that all three cues were informative about estimation accuracy, but only the number cue has a significant utility for optimal revision whereas the utilities of the other cues are substantially smaller compared to that of the number cue. Thus, for effective social revision of SP, participants were supposed to increase the degree of accepting others' SP estimates as the number of other people increases without much considering the other cues ('confidence' and 'agreement'), which have only a slight or negligible utility compared to the number cue. Nevertheless, people changed the degree of acceptance not just based on the number cue, but also based on the agreement between others and their own confidence in a degree similar to that for the number cue.

## Results

### Revision of initial SP estimates after observing others' SP estimates

To adapt a real-world problem of learning SP to a laboratory setup, we presented the following scenario to participants: "You are going to see 24 visual artworks, which have been exhibited at an online gallery for a month. More than one thousand people have visited the gallery and expressed their preferences for the pieces of art by clicking on either a 'like' or 'dislike' button for each item. By subtracting the number of 'dislike' marks from the number of 'like' marks for each item, we ranked the artworks in the order of popularity by assigning '1' to the most preferred item and '24' to the least preferred item." Hereafter, for the sake of simplicity, we will describe the scenario and name the variables in the first-person standpoint, such that '*my*' is used to indicate a participant who was asked to revise the initial estimates and 'others' is used

to indicate the other participants whose estimates were shown to that participant before revision.

Our experiment consisted of three steps. In the first step (Fig 1A), we showed participants the artworks one by one and asked them to estimate the rank of each item ($\widehat{SP}^m$, *my* SP estimate), and to express their estimated ranking and confidence ($c^m$, *my* confidence) by betting virtual coins on the rankings around the most probable (estimated) ranking of the item (see Materials and Methods for a detailed description). Here, $c^m$ was expected to vary from item to item and was thus considered as a potential cue that could carry useful information about the accuracy of $\widehat{SP}^m$, i.e., how close $\widehat{SP}^m$ was to the actual value of *SP*.

In the second step, we showed participants others' SP estimates of the same items in the first step while varying the number of other participants (Fig 1B). Others' estimates can be summarized by four descriptive statistics, as follows: (i) the average of SP estimates ($\widehat{SP}^o$, 'SP estimate of others'); (ii) the number of SP estimates ($n° \in \{1,2,3\}$); (iii) the reciprocal of the standard deviation of SP estimates ($a°$, 'agreement in SP estimates between others'); and (iv) the average confidence of others ($c°$, 'confidence of others'). Here, the last three variables, $n°$, $a°$, and $c°$, which varied across items, were considered as potential cues that might carry useful information about the accuracy of the first variable, $\widehat{SP}^o$.

In the final step, we repeated the first step by asking participants to estimate the SPs of the same artworks again ($\widehat{SP}^r$, 'revised SP estimate') after having been exposed to the SP estimates of others (Fig 1C).

## A regression model for social revision of SP estimates

The above scenario can be summarized in the first-person standpoint of an individual as follows (Fig 1A–1C): "*I* formed *my* initial SP estimate for a given artwork ($\widehat{SP}^m$), was then 'socially' exposed to the SP estimates formed by a few other participants who saw the same artwork that *I* saw ($\widehat{SP}^o$), and am now about to revise *my* initial SP estimate into a new one ($\widehat{SP}^r$) by considering the 'social' SP estimates of the others ($\widehat{SP}^o$)." Reasonably enough, we assumed that $\widehat{SP}^r$ is a compromise between $\widehat{SP}^m$ and $\widehat{SP}^o$. What remains unknown is the modulatory contributions of the potential accuracy cues $c^m$, $n°$, $a°$, and $c°$ to the compromise between $\widehat{SP}^m$ and $\widehat{SP}^o$; this was the focus of the current study. For example, "Do *I* shift *my* own estimate more toward the others' estimates when feeling less confident than when more confident?" Or, "Do *I* shift *my* own estimate a bit less when others' estimates are less similar to one another (disagreed) than when more similar (agreed)?" To address such questions quantitatively, we formalized the concurrent, modulatory contributions of $c^m$, $n°$, $a°$, and $c°$ to the compromise between $\widehat{SP}^m$ and $\widehat{SP}^o$ using a regression model in which the regression of $\widehat{SP}^r$ onto $\widehat{SP}^m$ and $\widehat{SP}^o$ are moderated by $c^m$, $n°$, $a°$, and $c°$, as follows:

$$\widehat{SP}^r = \widehat{SP}^m + W\,(\widehat{SP}^o - \widehat{SP}^m) + B + \epsilon; \qquad\qquad (\text{Eq 1})$$

$$W = W_0 + \beta_{cm}c^m + \beta_n\,n^o + \beta_a\,a^o + \beta_{co}\,c^o$$

where the first equality proposition captures the regression of the revised estimate onto the compromise between *my* estimate and others' estimate with a regression coefficient *W*, whereas the second captures the moderation of *W* by the four moderators (the potential accuracy cues $c^m$, $n°$, $a°$, and $c°$) with the corresponding moderation coefficients $\beta_{cm}$, $\beta_n$, $\beta_a$, and $\beta_{co}$. B, an intercept term, was used to capture a constant shift from $\widehat{SP}^M$ to $\widehat{SP}^R$ that cannot be regressed onto $\widehat{SP}^o - \widehat{SP}^m$ (e.g., a tendency to revise the SP slightly in a positive direction, even when $\widehat{SP}^o$ is equal to $\widehat{SP}^m$). $W_0$, a coefficient term, reflects the baseline degree of accepting

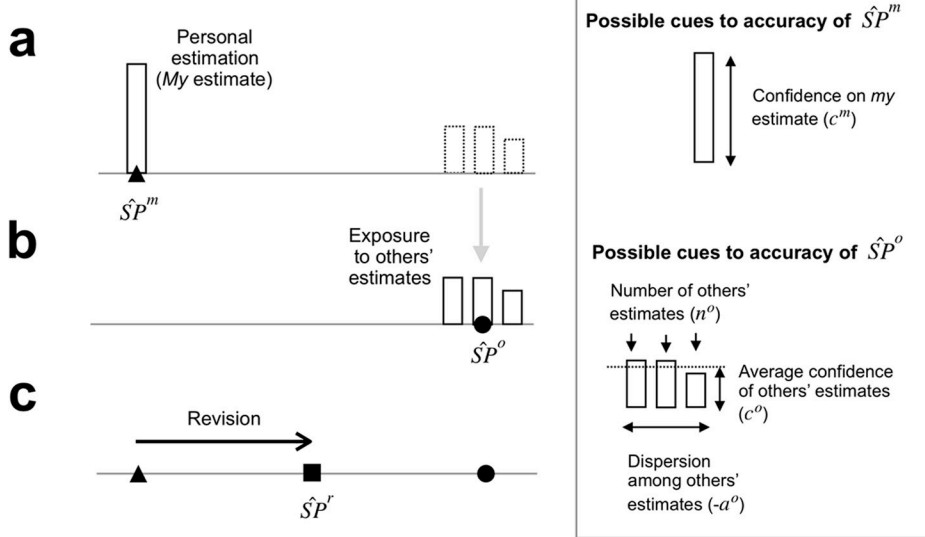

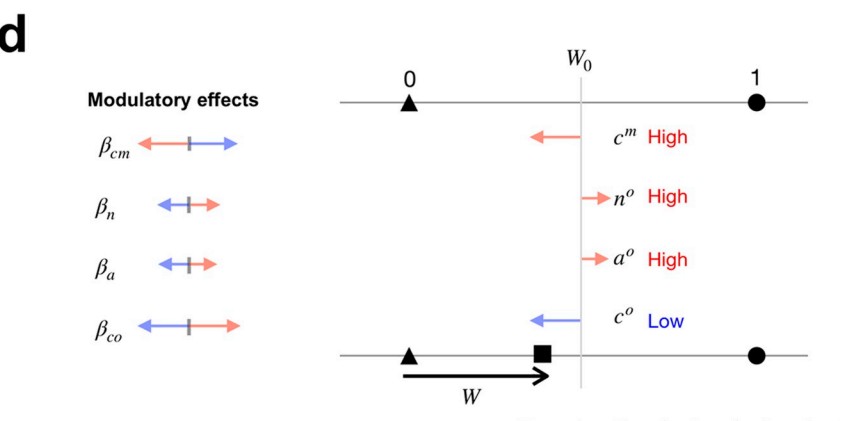

**Fig 1. Example illustrations of the SP revision task and possible modulatory contributions of accuracy cues to SP revision.** (**a-c**) The three steps of the task. (**a**) *My* SP estimation. On each trial, participants viewed a piece of visual artwork and estimated its rank, then betting coins on the estimated rank ($\widehat{SP}^m$, triangle; coins are also betted on nearby rankings but not shown here, see Materials and Methods). The height of the betted coins corresponded to the value of "*my* confidence" ($c^m$) on that estimated rank, and was a potential cue for "*my* SP estimation accuracy", as illustrated in the right-hand panel. (**b**) Exposure to others' estimates. Participants were exposed to the initial SP estimates that were made by a few selected others (three others in this example), along with the visual artwork. *My* estimate was not shown, and the mean of others' estimates (circle) was considered as "others' estimate" ($\widehat{SP}^o$) in the regression model. As illustrated in the right-hand panel, the number of others' estimates, reciprocal spread among their estimates, and average betting on the estimates correspond to the values of "number of others" ($n°$), "agreement" ($a°$), and "others' confidence" ($c°$), respectively, each of which is a potential cue for the accuracy of mean of others' SP estimation. (**c**) SP revision. After being exposed to others' estimates, participants were given a second chance to rate the SP of the artwork again ($\widehat{SP}^r$, square). As represented by the direction and length of the arrow, SP revision (the square) can be conceptualized by how much participants shifted their initial SP estimate (triangle) toward the SP estimates of others (circle). (**d**) Schematic illustration of the modulatory effects of the accuracy cues on SP revision according to a regression model. Using the regression model described in the text (Eq 1), we quantified the degree of accepting others' SP estimates with the regression coefficient $W$, which is shown by the black arrow. Importantly, we quantified the modulatory effects of the accuracy cues ($c^m$, $n°$, $a°$, $c°$) with the coefficients ($\beta_{cm}$, $\beta_n$, $\beta_a$, $\beta_{co}$) of the regression model; sum of the modulatory effects ($\beta_{cm}c^m + \beta_n n° + \beta_a a° + \beta_{co}c°$) comprise $W$ by adding to $W_0$, which represents the mean degree of acceptance across trials. The colored arrows show a hypothetical set of modulatory effects of the accuracy cues for the example shown in (a-c). In this example, participants modulated the degree of acceptance less than the average ($W < W_0$) because the total sum of the effects of moderators was negative.

others' SP estimates when the moderators on $W$ are set to zero. $\varepsilon$ is a noise term of the regression model. Superscripts indicate whether the given terms (SP estimates or accuracy cues) are associated with *my* ($m$) or others' ($o$) estimates.

This regression model allowed us to intuitively interpret the coefficients (Fig 1D). The value of $W$ can be considered as the "degree of acceptance", whereby values of 0 and 1 represent "*I* retain *my* initial opinion and do not consider the opinions of others at all" and "*I* completely accept the opinions of others while discarding *my* own opinion", respectively. Accordingly, the $\beta$ value can be considered as the extent to which a given moderator modulates the degree of accepting the opinions of others (the value of $W$). Also, given that the variable $\widehat{SP}^o$ is the mean of others' SP estimates (when the number of others is more than one), $W$ can be interpreted as a degree of conformity to the average of others' estimates.

Using this regression model, we evaluated the concurrent contributions of *my* subjective confidence ($c^m$), the number of estimates from others ($n°$), the agreement of estimates between others ($a°$), and the subjective confidence of others ($c°$) to the compromise between the *my* SP estimate ($\widehat{SP}^m$) and others' SP estimates ($\widehat{SP}^o$). First, we evaluated the predictive validity of the four candidate cues for SP estimation accuracy by examining whether their trial-to-trial values could predict the accuracy of trial-to-trial *my* (or others') SP estimates. More specifically, we tested whether $c^m_{ij}$ contained information about the accuracy of $\widehat{SP}^m_{ij}$ (where $i$ indicates the individual and $j$ indicates the identity of an item) and whether $n°_{ij}$, $a°_{ij}$, and $c°_{ij}$ contained information about the accuracy of $\widehat{SP}^o_{ij}$. We found that all four cues carried moderate but significant degrees of accuracy information. In other words, they could predict how close the SP estimate for a given item would be to the actual SP of that item. Next, having confirmed the predictive validity of the accuracy cues, we checked whether the accuracy cues have 'utility' and quantified the ideal way of using those cues for SP revision by conducting an 'ideal-learner' analysis. This analysis computes the values of the moderation coefficients that should be assigned to those accuracy cues to benefit maximally from the social revision of SP estimates, i.e., to maximize the resultant accuracy of the revised SP estimates ($\widehat{SP}^r_{ij}$). These 'ideal' values of $\beta_{cm}$, $\beta_n$, $\beta_a$, and $\beta_{co}$ allowed us to interpret the utility and ideal use of the accuracy cues in two important ways, as follows: (i) qualitatively, any non-zero ideal values of $\beta$ indicate that a given accuracy cue has 'utility' because it means that the revised estimate will be more accurate if the degree of acceptance is modulated by the accuracy cue ($\beta \neq 0$) than if not ($\beta = 0$); (ii) quantitatively, a specific ideal value of $\beta$ sets the 'criterion' for the most effective usage of a given accuracy cue, such that any deviation from the ideal value of $\beta$ implies a suboptimal use of that accuracy cue. As a final step, we characterized how people used the accuracy cues by fitting the regression model to their actual social revision of SP estimates, and evaluated how effectively people used the accuracy cues by comparing the fitted values of $\beta_{cm}$, $\beta_n$, $\beta_a$, and $\beta_{co}$ with their respective ideal criteria.

## Predictive validity of the cues for SP estimation accuracy

To evaluate the predictive validity of the four candidate cues for SP estimation accuracy, we needed to define the actual values of SP and an error metric for SP estimation. Participants were informed that the actual SP values had been determined by the preferences expressed by many people, who had not participated in the same experiment. In line with the instruction, we ran an online survey from an independent population of participants to acquire the average preference score for each item (See Materials and Methods for details). And based on those scores, we determined the ranking scores of the items in each genre, and took those as the actual SP values of the items. Next, to quantify the accuracy of an SP estimate, we calculated

the absolute amount of its deviance from the actual SP value ($SP_j$) as follows: $D^m_{ij} = |\widehat{SP}^m_{ij} - SP_j|$ for *my* estimates and $D^o_{ij} = |\widehat{SP}^o_{ij} - SP_j|$ for the averages of others' estimates (Fig 2A). Previous studies have primarily used this deviance value, often dubbed "absolute error", as the error metric for social revision of opinions [23,24,30,31]. Squared errors can be considered as an alternative error metric, but we opted not to use them because they are sensitive to a small number of extreme values [32]. Simply put, the smaller a deviance is, the more accurate an SP estimate is, which means that the value of a given SP estimate, either of *mine* ($\widehat{SP}^m_{ij}$) or of others ($\widehat{SP}^o_{ij}$), becomes closer to the actual value of SP estimate ($SP_j$). To avoid confusion, we will denote the deviance measures for *my* and others' SP estimates as $D^m_{ij}$ and $D^o_{ij}$, respectively. However, for the sake of simplicity, $i$(individual) and $j$ (item) will be omitted when they are obvious from the context.

Having defined the actual value of SP ($SP$) for each item and the errors for SP estimates ($D^m$ and $D^o$) on each trial, we evaluated the predictive validity of the accuracy cues by testing whether $D^m$ is dependent on the values of $c^m$ and whether $D^o$ is dependent on the values of $n^o$, $a^o$, and $c^o$.

We evaluated the dependency of the errors of *my* SP estimates ($D^m$) on *my* confidence ($c^m$) in the following procedure. First, within each individual, we split all SP estimates into two groups depending on whether the values of $c^m$ were lower or higher than their median value. This 'median split' discretization of $c^m$ within each participant was adopted to normalize the raw values of $c^m$ because the psychological scaling of subjective confidence is likely to be non-linear [29]. Second, we pooled together the $D^m$ values across participants within each of the two groups of discretized binary $c^m$ values and contrasted the pooled distributions of $D^m$ values between the two groups by plotting their complementary cumulative distribution functions (1-CDF; Fig 2B). The complementary CDF of a variable allowed us not only to visualize how a variable was geometrically distributed, but also to summarize its expected value using the area under the curve (AUC) [33]. Visual comparison of the AUCs between the two confidence conditions revealed that the mean deviance of *my* SP estimates from actual SP values was significantly larger when *my* confidence was low ($E(D^m|c^m = low) = 5.34$, 95% CI = [5.23, 5.46]) than when it was high ($E(D^m|c^m = high) = 5.08$, 95% CI = [4.95, 5.20]; one-sided permutation test, $p < 0.01$; Fig 2D). This indicates that the changes in accuracy of *my* SP estimation across trials can be predicted by *my* confidence.

The procedure for evaluating the predictive validity of the accuracy cues for others' SP estimation ($n^o$, $a^o$, and $c^o$) was identical to that of *my* confidence ($c^m$) for *my* SP estimation, except for the first step, whereby the raw measures of accuracy cues were differently preprocessed. As for the number of others ($n^o$), the original three discrete values (1, 2, and 3) were used without any transformation. The expected value of $D^o$ significantly decreased as the number of others increased ($E(D^o|n^o = 1) = 5.02$, 95% CI = [5.07, 5.36]; $E(D^o|n^o = 2) = 4.58$, 95% CI = [4.46, 4.70]; $E(D^o|n^o = 3) = 4.25$, 95% CI = [4.12, 4.36]; one-sided permutation tests for each pair, $p < 0.001$ for all possible pairs; Fig 2E). As for the agreement between others ($a^o$), we normalized the raw measures of $a^o_{ij}$ by categorizing them into high(er than the median) and low(er than the median) values within each of the two respective pools of $a^o_{ij}$. This normalization was applied separately for the trials in which the number of others ($n^o$) was 2 or 3 to orthogonalize the levels of $a^o$ against the levels of $n^o$ so that the predictive validity of $a^o$ for the accuracy of others' SP estimation could be evaluated independently of the levels of $n^o$. The expected value of $D^o$ was significantly smaller when conditioned on the high level of $a^o$ ($E(D^o|a^o = high) = 3.98$, 95% CI = [3.85, 4.09]) than on the low level of $a^o$ ($E(D^o|a^o = low) = 4.87$, 95% CI = [4.76, 4.99]; one-sided permutation test, $p < 0.001$; Fig 2E). As for others' confidence ($c^o$), the same preprocessing procedure was applied as for *my* confidence ($c^m$), i.e., the raw values of $c^o$ were

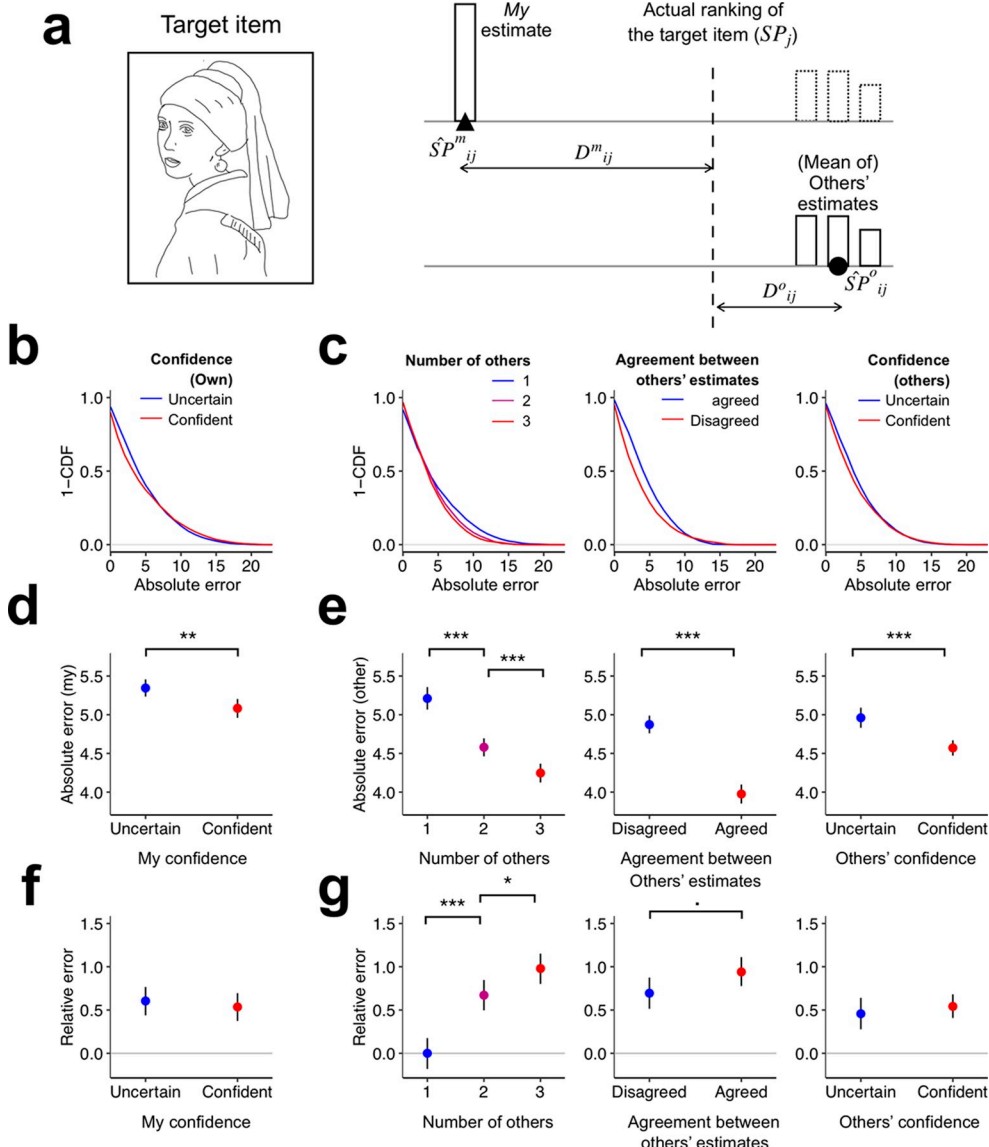

**Fig 2. Validity and utility of the accuracy cues.** (**a**) Error metric for SP estimation. For a given individual $i$, two estimation errors were defined for a given item $j$, one quantifying the absolute deviance of *my* (i.e., individual $i$'s) SP estimate from the actual SP value of the item ($D_{ij}^m = |\widehat{SP}_{ij}^m - SP_j|$), and the other quantifying the absolute deviance of others' SP estimate from the actual SP value of the item ($D_{ij}^o = |\widehat{SP}_{ij}^o - SP_j|$). (**b, c**) Complementary CDFs of absolute estimation errors according to accuracy cue levels. (**b**) Complementary CDFs of the absolute errors of *my* estimates ($D^m$) for the low (blue) and high (red) levels of *my* confidence ($c^m$). (**c**) Complementary CDFs of the absolute errors of others' estimates ($D^o$) for the different levels of number of others ($n^o$; left panel), agreement between others ($a^o$; middle panel), and others' confidence ($c^o$; right panel). (**d, e**) Means of absolute estimation errors for accuracy cue levels. Error bars show the 95% confidence interval of means according to non-parametric bootstrapping. (**d**) Means of the absolute errors of *my* estimates for low (blue) and high (red) levels of *my* confidence. (**e**) Means of the absolute errors of others' estimates for the different levels of the number of others ($n^o$; left panel), agreement between others ($a^o$; middle panel), and others' confidence ($c^o$; right panel) (**f, g**) Means of relative estimation errors for accuracy cue levels. Error bars show the 95% confidence interval of means according to non-parametric bootstrapping. (**f**) Means of the relative errors of others' estimates to *my* estimates ($D^{Rel}$) for the low (blue) and high (red) levels of *my* confidence. (**g**) Means of the relative errors of others' estimates to *my* estimates for the different levels of number of others ($n^o$; left panel), agreement between others ($a^o$; middle panel), and others' confidence ($c^o$; right panel). The small marks above the horizontal brackets indicate the significance of the difference between two data points (•, p<0.1; *, p<0.05; **, p< 0.01; ***, p<0.001). CDF: Cumulative density function.

categorized into high(er than the median) and low(er than the median) values within each individual. The expected value of $D^\circ$ was significantly smaller when conditioned on a high level of $c^\circ$ ($E(D^\circ|c^\circ = high)$ = 4.57, 95% CI = [4.47, 4.67]) than on a low level of $c^\circ$ ($E(D^\circ|c^\circ = low)$ = 4.96, 95% CI = [4.83, 5.09]; one-sided permutation test, p < 0.001; Fig 2E).

In sum, all of the candidate accuracy cues significantly predicted the deviance of *my* SP estimation ($c^m$) or others' SP estimation ($n^\circ$, $a^\circ$, and $c^\circ$) from the actual SP values. In other words, the four accuracy cues were all informative about the accuracy of given SP estimates.

## Utility of the cues for SP revision

As mentioned earlier, a given accuracy cue can be said to have 'utility' only when people can benefit from appropriately modulating the degree of accepting others' SP estimates across trials based on that cue. While the results described above (Fig 2B–2E) indicate that the cues carry predictive information about the accuracy of SP estimates themselves, this does not necessarily imply that those cues have 'utility' for SP revision. The utility of any given cue in SP revision can only be guaranteed when the cue carries information about the 'relative accuracy' of others' SP estimates ($\widehat{SP}^o$) compared with *my* SP estimates ($\widehat{SP}^m$) across trials. We can imagine some hypothetical situations in which a certain cue has validity for SP estimation accuracy but has no utility for SP revision. In one such situation, a spurious association between an accuracy cue and estimation accuracy may arise from the across-item variability in difficulty. For example, when a given item is easy for estimation, *my* estimation and confidence tend to be accurate and high, respectively. In turn, this will mean *my* confidence is capable of predicting the accuracy of *my* estimation. However, others' estimation accuracy also tends to be high for an easy item. As a result, *my* confidence is related to the accuracy of both *my* estimation and others' estimation, but may not carry any information about the relative accuracy between *my* estimation and others' estimation. As shown in this example, the fact that a given cue can predict the accuracy of *my* or others' SP estimates does not guarantee that the same cue can predict the relative accuracy between *my* and others' SP estimates. For this reason, we assessed the utility of the accuracy cues by evaluating whether they can predict the relative accuracy between *my* and others' SP estimates across trials. To do so, we defined an additional error metric that reflects 'relative accuracy', $D^{Rel} = D^m - D^\circ$, which quantifies the proximity of others' SP estimate to the actual SP relative to the proximity of *my* SP estimate to the actual SP. In other words, this value indicates 'how other's estimate(s) is(are) more accurate than *my* estimate when others and *I* work on the same item.' Thus, if this value is high, we should accept others' estimates to a high degree.

Except for the number cue, the remaining three accuracy cues all failed to significantly predict $D^{Rel}$ (Statistical significance was evaluated using the one-sided permutation test; see Materials and Methods for details). The relative accuracy was not significantly higher when *my* confidence was lower than when it was higher ($E(D^{Rel}| c^m = low)$ = 0.642, 95% CI = [0.438, 0.766]; ($E(D^{Rel} |c^m = high)$ = 0.536, 95% CI = [0.373, 0.695]; one-sided permutation test, p > 0.1; Fig 2F); also others' confidence level was not significantly related to relative accuracy. ($E(D^{Rel} |c^o = low)$ = 0.457, 95% CI = [0.277, 0.640]; $E(D^{Rel} |c^o = high)$ = 0.542, 95% CI = [0.407, 0.68]; one-sided permutation test, p > 0.05; Fig 2G). The agreement between others predicted relative accuracy to a moderate extent ($E(D^{Rel} |a^o = low)$ = 0.694, 95% CI = [0.514, 0.876]; $E(D^{Rel} |a^o = high)$ = 0.940, 95% CI = [0.777, 1.11]; one-sided permutation test, p < 0.1; Fig 2G).

Unlike the other cues, relative accuracy significantly increased as the number of others increased ($E(D^{Rel} |n^o = 1)$ = 0, 95% CI = [-0.180, 0.176]; $E(D^{Rel} |n^o = 2)$ = 0.671, 95% CI = [0.496, 0.846], $E(D^{Rel} |n^o = 3)$ = 0.980, 95% CI = [0.802, 1.15]; one-sided pairwise permutation

tests, $p < 0.001$ for n = 1 > n = 2 and for n = 1 > n = 3; $p < 0.05$ for n = 2 > n = 3; Fig 2G). To summarize the results so far, whereas the four accuracy cues were all significantly informative about the accuracy of SP estimation, this informativeness was not so much useful in SP revision per se for the confidence and agreement cue as for the number cue.

## Ideal-learner analysis

Next, we carried out an ideal-learner analysis to define the ideal way of using the accuracy cues to modulate the degree of accepting others' SP estimates. Here, the ideal way refers to one that maximizes the benefit of social revision of SP estimation, i.e., one that minimizes the absolute deviation of revised SP estimates from actual SP values. We defined the ideal way of using the accuracy cues by finding the 'ideal' regression coefficients for the four moderators $c^m$, $n°$, $a°$, and $c°$ in the regression model described previously (Eq 1). To simplify interpretation and comparison of their coefficient values, we assigned the following dummy values to the moderators' categorical states: -0.5 to 'lower half', +0.5 to 'higher half', and 0 to 'median' for $c^m$, $a°$ and $c°$; -1 To fairly compare the effects of modulators in magnitude, we assigned the dummy variables to $n°$ such that they reflects the variance in a way comparable to that of the other modulators: -0.5 to 'n = 1', 0 to 'n = 2', and 0.5 to 'n = 3' for $n°$ (see Materials and Methods for details).

To find the ideal regression coefficients, we first created an ideal agent (instead of an actual participant) who 'ideally translates the observed values of *my* SP estimation ($\widehat{SP}^m_{ij}$) and others' SP estimation ($\widehat{SP}^o_{ij}$) into revised SP estimates (*ideal* $\widehat{SP}^r_{ij}$) by strictly following our revision model without any random noise in behavior ($\varepsilon = 0$). Note that we refer to the outcomes of this ideal revision as *ideal* $\widehat{SP}^r_{ij}$ to distinguish it from those of human participants ($\widehat{SP}^r_{ij}$). Second, we identified the ideal set of model parameters ($W_0^*$, $\beta_{cm}^*$, $\beta_n^*$, $\beta_a^*$, $\beta_{co}^*$, and $B^*$) that would minimize the absolute deviation of the revised SP estimates of the ideal agent (*ideal* $\widehat{SP}^r_{ij}$) from the actual SP values ($SP_j$) as follows:

$$[W_0^*, \ \beta_{cm}^*, \ \beta_n^*, \ \beta_a^*, \ \beta_{co}^*, B^*] \ = \ Argmin \sum_{i,j} \left| SP_j - ideal \ \widehat{SP}^r_{ij} \right| \tag{Eq 2}$$

where $i$ and $j$ indicate an individual and an item, respectively, which means that the data acquired from all experimental trials (n = 6,840) contributed to the cost function. Finally, we interpreted the coefficient parameters for the four moderators in Eq 2 ($\beta_{cm}^*$, $\beta_n^*$, $\beta_a^*$, and $\beta_{co}^*$) as the 'ideal' leverages that should be assigned to the four accuracy cues ($c^m$, $n°$, $a°$, and $c°$), respectively, to maximize the benefit of social revision of SP estimation.

The directions of the ideal leverage values were consistent with the utility results, whereby a significant positive shift was found for the number of others ($\beta_n^* = 0.257$), but the degree of shifts were quite smaller for the other cues than for the number cue: others' agreement ($\beta_a^* = 0.071$), own confidence ($\beta_{cm}^* = -0.049$) and others' confidence ($\beta_{co}^* = 0.026$) (Fig 3A). These results indicate that a rational agent who aims to maximally benefit from SP revision should modulate the degree of accepting others' SP estimates based on the accuracy cues as instructed by the directions and magnitudes of the ideal leverage values. Note that the leverage value assigned to the number cue is substantively greater than those assigned to the remaining cues, which is consistent with the results of the utility analysis. This means that the ideal learner should modulate the degree of acceptance mostly based on the number of the others but not much so based on the remaining cues. In what follows, we used these ideal leverage values as the ideal criteria for evaluating how effectively participants use the accuracy cues in SP revision.

## Human use of the cues for SP revision

As the final step, we evaluated how effectively people use the accuracy cues for SP revision by inspecting how closely human use of the accuracy cues followed the ideal. This was assessed using the ideal-learner analysis described in the previous section. Basically, to capture how participants modulated the degree of accepting others' SP estimates based on the accuracy cues, we used the same regression model (Eq 1) as was applied in the ideal-learner analysis. However, we introduced the following additional assumptions to address noisy responses and individual differences that were present in the data. First, the noise term $\varepsilon$, which was set to zero in the ideal-learner analysis, was assumed to be non-zero because, unlike the ideal learner, a human learner is likely to exhibit random variability in revision behavior (e.g., revising differently when the same trial is repeated). Specifically, we assumed that such noise would be distributed across trials in a beta binomial distribution rather than in a Gaussian distribution, which is typically assumed in conventional regression models. The beta binomial distribution is appropriate for the current study because the response variable (ranking scores) is bounded between 1 and 24, and because the error distribution shape tended to be skewed around the boundaries (see S1 Text for details). Second, to take into account the idiosyncratic differences in coefficient and responses bias across individuals, we built a hierarchical regression model (HRM; See S1 Text for details) by extending the basic regression model such that the coefficients of individuals were assumed to be random samples from a normal distribution with a mean, called a 'fixed effect'. The fixed effects in the HRM allowed us to quantify the typical way in which human participants, on average, use the accuracy cues to modulate the degree of accepting others' SP estimates.

The modulatory coefficients for the accuracy cues, which were estimated as the fixed effects in the HRM using a Bayesian estimation method [34] (see S1 Text for details), were as follows: $\beta_{cm} = -0.103$, $\beta_{co} = -0.012$, $\beta_a = 0.150$, and $\beta_n = 0.168$ (Fig 3A). The estimated coefficients all significantly deviated from zero, except for the subjective confidence of others; the 95% CI of the coefficient only included zero for $\beta_{co}$ ([−0.041,0.055]), and not for $\beta_{cm}$([−0.147,−0.055]), $\beta_n$ ([0.045,0.120]), or $\beta_a$ ([0.093,0.201]). When compared with the ideal learner's behavior, participants' actual usage of the accuracy cues was different in several aspects (Fig 3A). First, people used the number cue in the right direction, but the degree of acceptance modulation fell short to reach the ideal one. Second, people over-modulated the degree of acceptance based on the agreement cue and their own confidence cue compared to their respective ideal degrees of modulation. The only good correspondence between the ideal learners and human participants was found when using the cue of others' confidence: neither the ideal learner nor human participants modulated the degree of acceptance based on the others' confidence cue.

We also found that the coefficient for the overall acceptance of others' SP estimate ($W_0 = 0.49$, 95% CI [0.490, 0.532]) was significantly smaller than its ideal value ($W_0^* = 0.63$). $W_0$ would be close to 2/3, since on average, participants were exposed to two others' estimates, if participants gave equal weights to all estimates, including their own and individual estimates of others. However, 0.49 was significantly smaller than 2/3. This result is consistent with the well-known egocentric bias in opinion revision, whereby we sub-optimally give a greater weight to our own opinions than to those of others [16,17,35].

## Replication of main findings

When using alternative methods to analyze our data, we found that the main findings were not qualitatively affected. First, we estimated the modulatory effects of accuracy cues by fitting the basic regression model (Eq 1) with the deviation minimization method (Eq 2) for the ideal learner, and by fitting the HRM (see S1 Text for details) with the Bayesian estimation method

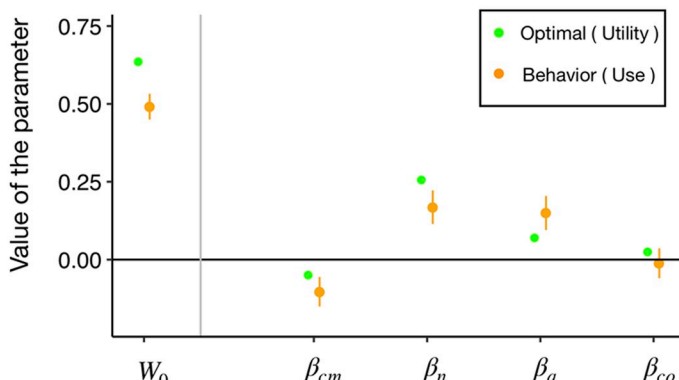

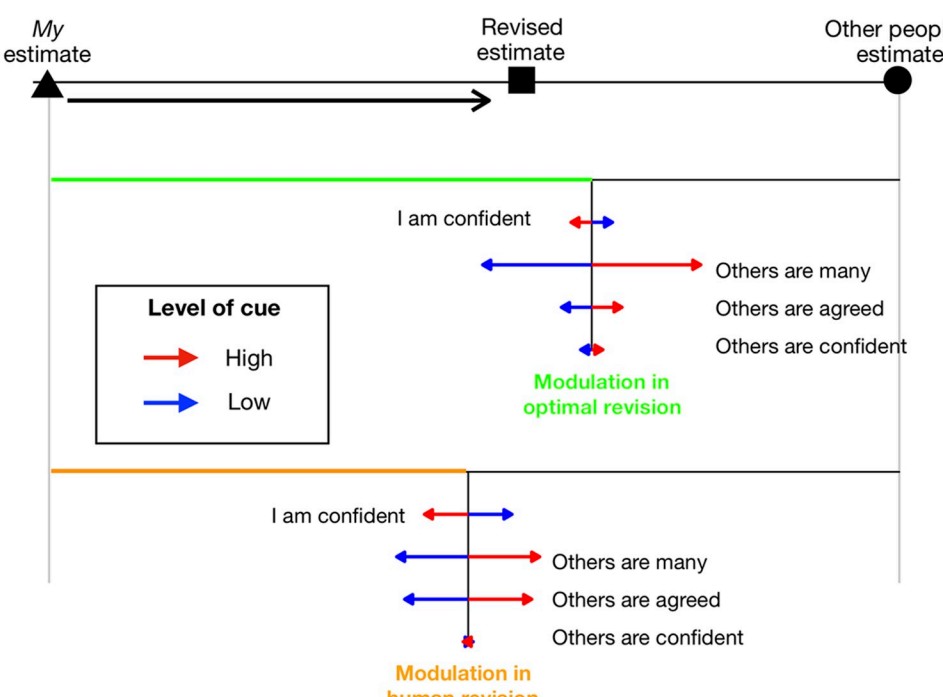

**Fig 3. Comparison between ideal and human SP revisions.** (**a**) Results of the regression analyses. Green and orange circles represent the model parameters estimated by the ideal-learner analysis (Eq 2) and the human data analysis (See S1 Text for details). Error bars are the 95% confidence intervals of the parameters, which were estimated by a Bayesian method (95% interval of posterior distribution of each parameter; see S1 Text for details). (**b**) An illustrative summary of the comparison between ideal and human SP revisions. Top, we conceptualized SP revision as the extent to which people accept others' SP estimate (black circle) by shifting their initial SP estimate (black triangle) to somewhere in between the initial and others' estimates (black square). Middle, using the ideal-learner analysis, we identified the optimal way of revising SP by quantifying the overall shift ($W_0^*$; length of green line) and the direction and magnitude of acceptance modulation for each of the accuracy cues $\beta_{cm}^*$, $\beta_n^*$, $\beta_a^*$, and $\beta_{co}^*$; red and blue arrows). Bottom, the human data analysis revealed that the human way of revising SP estimates is different from the ideal way of using the accuracy cues in social revision of SP estimates. Especially, the extent to which human individuals use their own confidence cue and the agreement cue much exceeded their respective ideal extents. Also, people showed a smaller degree of overall acceptance (orange horizontal line) than the optimal degree (green horizontal line).

for the human participants. Although the HRM is simply an extended version of the basic model that incorporates noisy responses and individual differences, it is possible that the observed mismatches between the ideal and human coefficients were a result of the use of different regression models and estimation methods. To address this possibility, we estimated the modulatory effects of accuracy cues for the human participants using the same equation and procedure as those used in the ideal-learner analysis. Specifically, we replaced the actual SP value ($SP$) with the observed, revised SP estimates of human participants ($\widehat{SP}^r$) in Eq 2 to find the modulatory coefficients that would be used by a hypothetical, noise-free learner whose SP revision behavior best matched the human SP revision. We refer to this alternative way of assessing human use of the accuracy cues as the 'noise-free human-learner analysis' (see S2 Text for details). Thus, the coefficients in the noise-free human-learner analysis can be considered as capturing the representative human use of accuracy cues without taking noisy responses and individual differences into account. The conclusion drawn from the coefficients acquired with the noise-free human-learner analysis did not qualitatively differ from our original conclusion that was based on the coefficients acquired with the HRM and Bayesian estimation method (see S2 Text).

Second, in the current regression models, we converted the raw values ({1, 2, 3}) of the accuracy cue of 'the number of others' ($n°$) into {-0.5,0,0.5} and evaluated its influence with a single coefficient ($\beta_n$). Although this procedure was adopted to be consistent with the procedure in which the other accuracy cues were converted and evaluated, it cannot reflect a potential nonlinearity in influence that might occur as the number of others increases. Specifically, a greater change in influence is expected when the number of others increases from 1 to 2 than when it increases from 2 to 3, according to previous work [23]. To address this nonlinearity, we modified the regression models such that the modulatory effects of the change from 2 to 1 and of that from 2 to 3 others were separately captured by two different regressor variables (see S2 Text). When these modified regression models were applied, the modulatory effect of the change in number from 2 to 1 was indeed greater than that from 2 to 3, both in the ideal-learner analysis and in the human use analysis. More importantly, the main findings concerning the utility and use of the remaining accuracy cues ($\beta_{cm}$, $\beta_n$, and $\beta_a$) were unaffected by use of the alternative regression models.

Finally, previous studies on advice-taking have often descriptively quantified the acceptance of others' advice using an index called "weight on advice" (WOA) [17,36], which measures a scaled shift of an individual's opinion toward others' advice on a trial-to-trial basis ((final estimate–initial estimate)/(advisor recommendation–initial estimate)). Thus, a WOA measure can be considered as corresponding to the coefficient $W$ in our regression models. By treating WOA as $W$, we evaluated the influence of the accuracy cues on WOA measures by examining whether the mean WOAs differed between the trial groups corresponding to the high and low levels of each of the accuracy cues, then, by regressing the WOA measures onto the accuracy cues in a multiple linear regression model. The results from both analyses supported our main findings: participants used '*my* own confidence', 'agreement between others', and 'the number of others', but did not use 'others' confidence' in SP revision, as summarized in Fig 3A (see S4 Text).

## Discussion

The current study investigated i) whether the candidate accuracy cues (number, agreement, and confidence) have 'utility' for optimal revision of opinions on problems with 'socially correct' answers and ii) whether people effectively use those cues in such revisions. As for the first question, we found that, while all those cues validly predicted the accuracy of SP estimates,

only the number cue had a significant utility for SP revision, but the remaining cues had only moderate or negligible degrees of utility compared to the number cue. So, for optimal revision, the degree of acceptance should be modulated mainly by the number of others, but only moderately or almost none by the other remaining cues. As for the second question, we found that people used the number cue in a manner consistent with the optimal way. But we found that people also used both of the agreement cue and own confidence cue significantly during social revision to the extent to which they used the number cue, despite the fact that the utilities of the agreement and confidence cues were a lot less than the utility of the number cue.

Our contribution is novel in that the 'number', 'agreement', and 'confidence' cues were concurrently evaluated for their utility and use in social revision for problems with socially correct answers, such as SP. Our contribution is also important in that social revision is considered as the most typical, if not the only, way to improve the accuracy of personally formed estimates on problems with socially correct answers, whereas there are many methods other than social revision that can be used to improve accuracy for problems with factually correct answers (e.g., acquiring additional information by referring to non-human sources). In what follows, for each of the accuracy cues, we will compare our findings with previous ones, and discuss how the utility and use of accuracy cues differ between different types of problem.

## Utility and use of the number-of-estimates cue

Our findings on the utility of the number cue are consistent with previous findings on other types of problem [23,25,37], both qualitatively and quantitatively. Qualitatively, the relative accuracy of the mean of others' SP estimates to individual's initial SP estimate increased as the number of estimates increased. Quantitatively, the magnitude of modulating the acceptance of others' SP estimates was greater when the number changed from 2 to 1 than when it changed from 2 to 3 (S2 Text). That difference in magnitude found in our study is consistent with previous findings, in that it closely followed the well-known ideal revision rule in which the optimal degree of accepting the averaged estimates of others (in our study, $\widehat{SP^o}$) is determined by $n/(n+1)$, where $n$ is the number of estimates. Likewise, our findings on the human use of the number cue are also consistent with those of previous studies [23,29,38], in that people decreased and increased the acceptance of others' SP estimates as the number decreased and increased, respectively. Put together, the number of opinions or estimates appear to be general accuracy cues that have utility and are used by people robustly, regardless of the types of problem.

However, the above conclusion about the effective human use of the number cue does not indicate that participants showed the optimal degree of accepting others' SP estimates across trials in an absolute sense. To make this claim, people must have both overall acceptance and modulatory effects that are matched to the ideal effects. However, according to our results, people showed significantly smaller overall acceptance ($W_0$ in our model) than the ideal, which suggests that they gave a significantly smaller weight to others' opinions (i.e. ego-centric discounting [11,16,35]) than the optimal weight. Thus, by "effective use" of the number cue for SP revision, we mean that people modulate the degree of accepting others' estimates in a relative sense, assigning a higher weight to others' estimates when the number is relatively larger despite the overall egocentric bias. In other words, if the egocentric bias can be overcome or mitigated, e.g., by taking other people's perspectives on board [39], people are potentially capable of regulating their SP revision based on the number cue in a near-optimal fashion in an absolute sense. This issue of distinguishing between the absolute optimality in weighting and the effective use of modulatory cues should also be applied to our interpretations about the human use of the other accuracy cues in the text below.

## Utility and use of the agreement-between-estimates cue

Compared to the number cue, it is less obvious whether the agreement cue has utility. It is possible that an agreement between people's estimates does not indicate the accuracy of those estimates but instead signals the presence of 'shared blind spots' that arise from shared experiences or prior interactions, which lead to shared errors [18,40–42]. On the other hand, it makes sense that an agreement between others' estimates is likely to indicate the accuracy of those estimates [43], probably because people tend to provide similar opinions when problems of interest are relatively easy, and thus those estimates are closer to the true answer [44]. However, if the association between the agreement cue and other's estimation accuracy is merely due to the across-item variability of problem difficulty, the agreement cue is also associated with my initial estimation accuracy to a similar degree. Then, the agreement cue becomes irrelevant to the accuracy of others' estimate relative to my estimate, which means that the association between estimation accuracy and the agreement cue is not sufficient for the agreement cue to have utility in social revision. Despite the marginal statistical significance (p~0.05), the agreement cue had a slight degree of utility in our data. Then, what possibly makes the agreement cue have utility in social revision?

From a statistical standpoint, the work by Yaniv and colleagues [24,25] has suggested that the shape of the distribution of individuals' estimates around the true value is critical for determining whether agreement between estimates has utility for social revision. In particular, the authors found that discounting opinions that deviate from the mean is an effective strategy for improving the accuracy of social revision when the distribution of estimates has thicker tails than would be expected under a normal distribution. It makes sense to use the strategy of deviant discounting because such distributions imply the prevalence of outlying estimates, which is known to occur frequently in various types of human estimation [26]. In our data, the kurtosis of error distribution was larger than three (3.2), which means that the error distributions had slightly thicker tales than would be expected under a normal distribution. For this reason, the agreement cue might have a slight—but statistically insignificant—degree of utility.

Despite its weak and insignificant degree of utility, people's use of the agreement cue in our task was very clear. The degree to which people accepted others' opinions was significantly affected by the agreement between others estimates, and the acceptance degree was comparable to that of the number cue. People's use of the agreement cue (e.g., deviant discounting) has been reported in many different kinds of tasks, including not just advice-taking tasks [24,25,45], but also those involving numerical or emotional integration [46,47]. In this regard, the current findings extend those previous findings by showing that people use the agreement cue for problems with socially correct answers. Here an interesting question is whether the degree of using the agreement cue (or deviant related behavior) varies across problem domains. We conjecture that people's use of the agreement cue could be more pronounced in our task because of the 'social' nature of SP estimation, which may prompt people to conform to 'socially agreed' estimates while avoiding 'socially deviant' estimates. Further work is required to investigate this possibility by, for example, comparing human use of agreement cues between problems with 'socially correct' answers and those with 'factually correct' answers.

## Utility and use of the subjective-confidence cue

Previous work has suggested that the utility and use of confidence cues are highly dependent on the type of problems. Both the utility and human use of confidence cues are evident in revising opinions for problems with 'objectively correct' answers, so called "intellective problems" (e.g., a mathematical question), but are very weak or negligible for problems with 'subjectively correct' answers, so called "judgmental problems" (e.g., rating a movie) [13,28]. On

the other hand, some studies have suggested that confidence cues carry valid information about the accuracy of an estimation for problems that involve 'social prediction' (e.g., predicting peers' responses to hypothetical daily events) [48,49]. We found that the confidence cue has significant information about estimation accuracy of SP estimation per se but has only a negligible degree of utility for social revision of SP estimates. This result indicates that people have the metacognitive ability of predicting whether their own estimation about 'how other people like a given item' is accurate or not, but that they are not supposed to decide to accept other's opinions based on their subjective feeling of confidence for optimal revision. More specifically put, an SP estimate with higher levels of confidence surely indicates that the estimate on a given item is more accurate than the estimates on other items with lower levels of confidence, but they do not warrant that their own SP estimates are more accurate than the other people's estimates about the same item.

The lack of confidence utility is different from previous reports, where confidence has utility during social revision for solving problems that have 'factually correct' answers [24,50]. We think that, in such kinds of problems, trial-to-trial changes in confidence have a stronger relationship with accuracy, and this relationship remains useful even when accuracy is defined in a relative manner. More specifically, when solving problems with factually correct answers, the trial-to-trial variability in confidence is likely to indicate how accurate an estimate on a given item is compared to the other people's estimates on the same item. But this seems to be not the case for solving problems with socially correct answers, as those used in our study.

One of the most intriguing findings of the current study is the asymmetry in the use of the confidence cue; although both individual's own and others' confidence have negligible degrees of utility for SP revision, people substantively modulated the acceptance of others' SP estimates based on their own confidence, but not at all based on others' confidence. To our knowledge, this is the first report of the asymmetric use of confidence for social revision. Previous advice-taking or group-decision-making studies have mainly focused on whether people use others' (e.g. advisor's) confidence to adjust the acceptance of others' advice or decisions [17,21,44,51]. One possible contributor to the asymmetric use of confidence found in our study could be the low degree of 'demonstrability or persuasibility' of problems with 'socially correct' answers [20]. According to this idea, problems differ along a spectrum in demonstrability, which is the extent to which people can persuasively demonstrate to others that their answer is the correct one, with intellective and judgmental problems at each end of spectrum. Problems with 'socially correct' answers, such as SP estimation (e.g., "Which movie do Korean people prefer most?") seem slightly more 'demonstratable' than judgmental problems (e.g., "Which movie is funnier?"), but a lot less 'demonstrable' than intellective problems (e.g., "What is the probability of rolling two fives in a row with one die?"). If so, 'confident' estimates for socially correct or judgmental problems are less acceptable than those for intellective problems. This idea is in line with previous reports that people modulate the acceptance of others' opinions based on others' confidence when dealing with intellective problems, but not so much when dealing with judgmental problems [13,28]. By contrast, we suggest that people modulate the acceptance of others' SP estimates based on their own confidence because there is no need for 'demonstration' or 'persuasion' about the accuracy of their own estimates.

## Limitations and future directions

Our results about the human use of accuracy cues are a population-level summary of how people, on average, use those cues for social revision, and do not address the issues of individual differences or inter-trial behavioral variability, which were treated as nuisance variables (random effects and noise) in our regression model (the HRM). Thus, our results do not

necessarily indicate that every person modulates the degree of accepting others' SP estimates based on the accuracy cues to the extents indicated by the modulatory regression coefficients ($\beta_{cm}$, $\beta_n$, $\beta_a$, and $\beta_{co}$) on every single trial. Our data suggest that individuals differ considerably in how consistent their use of the accuracy cues is with the ideal use, and that there is considerable trial-to-trial variability in modulatory behavior. While these interesting research topics were beyond the scope of the current work, they could be the subject of future work.

Another remaining issue is related to the fact that we asked people to estimate the ranking of items within a given set of items [31,52]. We opted to do so mainly because people's preferences for sociocultural items are typically expressed on an ordinal scale (e.g., "This song is ranked at number 3"). Furthermore, rank estimates are naturally normalized within each individual and thus can be easily shared between people. By contrast, other not-naturally-normalized estimates such as ratings are not readily sharable between people because the estimation criterion might differ between individuals; this makes it unclear as to whether observed differences in estimation are due to differences in item properties or differences in estimation criterion. Despite these merits, the ranking method we used has one drawback; due to its bounded nature, the extremity of estimates tends to covary with subjective confidence or agreement between others' estimates. In particular, estimates that are close to the extreme values tend to be accompanied by higher confidence measures than do estimates in an intermediate range. This link between confidence and extremity in estimation makes it difficult to determine whether the observed utility or use of a confidence cue should be ascribed to subjective confidence itself or, alternatively, to extremity in estimation. This issue limits our conclusions about the utility and use of the confidence cue for social revision and calls for the use of an alternative method of acquiring SP estimates other than the ranking method in future studies. Nonetheless, considering that SP values are prevalently ranked in our daily social life, and that confidence and extremity in bounded-scale estimation (in our case, ranking) are naturally coupled with each other [29,53], we believe that our conclusions about the utility and asymmetric use of confidence still contribute to our understanding of the contributions of subjective confidence to social revision of SP in real-world situations.

## Materials and methods

### Participants

A total of 80 undergraduate students at Seoul National University volunteered to participate in the 90-minute experiment in exchange for a compensation of 13 USD. Participants all had normal or corrected-to-normal vision and were recruited via the SONA system, which is a cloud-based participant management software (https://snucube.sona-systems.com/). Written informed consent was obtained from all participants before the experiment, which was performed in compliance with the safety guidelines for human experimental research, as approved by the Institutional Review Board of Seoul National University. Written informed consent was obtained from all participants before the experiment, which was performed in compliance with the safety guidelines for human experimental research, as approved by the Institutional Review Board of Seoul National University. (IRB No. 1604/003-014).

We carried out a total of 20 experimental sessions; in each session, four participants were asked to perform an experimental task as a group. All four volunteers showed up for 17 sessions, but one volunteer did not show up for the remaining three sessions. As a result, a total of 77 volunteers (36 female; aged 18–27 years; mean age, 21.7 years; standard deviation, 2.46 years) participated in the experiment, 68 as part of a four-person unit and 9 as part of a three-person unit. Although participants signed up for experimental sessions independently and at their convenience, there were sessions in which two participants happened to know each other

before participating in the experiment. Otherwise, none of the participants in the same unit had met before the experiment.

### Experimental setup and stimuli

Each experimental session was carried out in a quiet room. Four participants (and three participants for the exceptional three sessions as described earlier) sat facing each other around a round table (radius, 0.45 m) such that they could hear the key strokes made by their partners; this setup was intended to promote social presence (Fig 4B). However, the monitors and response boxes were arranged such that individual participants could not see their partners' responses.

On each trial, participants viewed a visual artwork on a computer screen (Fig 4A) and guessed how it would be ranked in popularity for a given genre of visual art by betting virtual coins on a certain range of ranks. They were told that they would only win coins invested at the actual rank (blue bar in Fig 4C). A monitor display (1280 x 1024 pixels at 60 Hz refresh rate, LG Flatron L1954TP_PF, LG Electronics, Nanjing, China) and a number pad were assigned to each participant. Each participant performed the task twice on each of the total 120 artworks; they initially guessed the rank without knowing what their partners had guessed (initial-round trials, Fig 4D), and then had a second chance to guess the rank after seeing their partners' guesses (revision-round trials, Fig 4D).

We chose artworks that varied considerably in subjective task difficulty so that betting responses would vary not only between artworks, but also between participants. We did so to generate sufficient trial-to-trial variability in subjective confidence and reduces the trial-to-trial co-variability between *my* confidence and others' confidence. If not, it would have been difficult to observe the effects of confidence due to its small variability, and difficult to differentiate the effects of *my* confidence and those of others' confidence due to the co-variability between the two variables. To create sufficient degrees of variability in difficulty across both artworks and participants, we selected artworks using the following procedure. First, 24 artworks from five genres of visual art were selected, namely, 'architecture', 'fashion design', 'industrial design', 'painting', and 'sculpture.' We chose these genres under the assumption that the participants (undergraduate students) would have varying degrees of knowledge about them. Second, we created variability in the task difficulty between artworks within a given genre. For this, we asked five graduate students who were majoring in fine art to recommend 25 to 60 items that would fit into one of three aesthetic categories ('negative', 'neutral', and 'positive' subjective feeling). We chose the final 24 artworks for each genre according to preference surveys completed by students who did not major in any arts, such that their averaged preferences were roughly evenly distributed.

### Task

At the beginning of each experimental session, we told participants the following: "You are going to see 24 visual artworks, which have been exhibited at an online gallery for a month. More than a thousand people have visited the gallery and expressed their preferences by clicking on either a 'like' or 'dislike' button for each item. By subtracting the number of 'dislike' marks from the number of 'like' marks for each item, we ranked the artworks in the order of popularity by assigning '1' to the most preferred item and '24' to the least preferred item. You will be asked to guess these rankings." Participants then completed five blocks of trials, one block for each genre. Each block comprised two rounds of trials, including 24 initial-round trials followed by 24 revision-round trials.

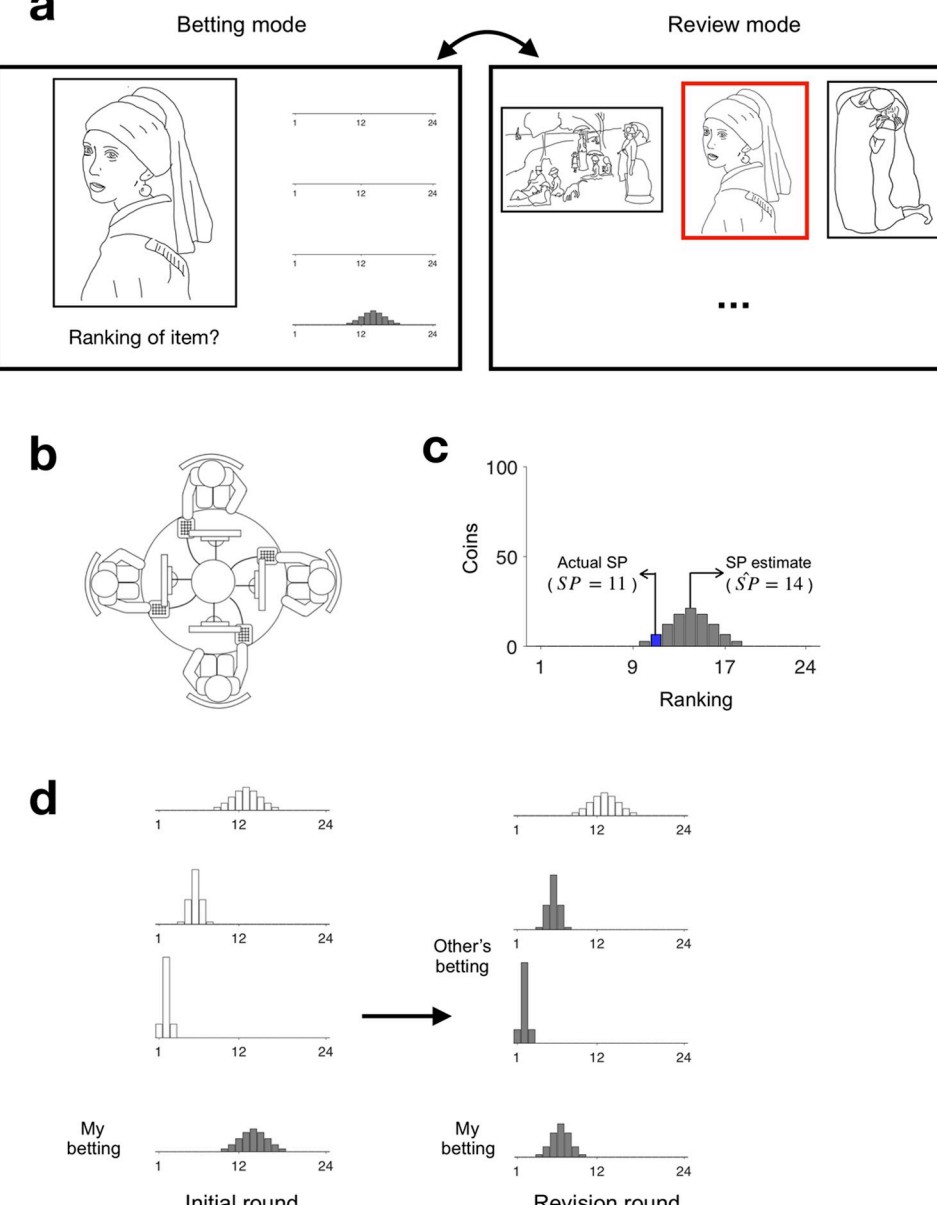

**Fig 4. Experimental setup, stimuli, and task.** (a) Two modes of stimulus view. Left ("betting mode"), only the target item on a current trial was shown with the betting axes. Participants expressed their bets about what the actual SP ranking of the item was on the axis at the bottom on both initial-round and revision-round trials, and viewed their partners' bets on the axes above their own on revision-round trials. Right ("review mode"), the target item (demarcated with a red box) was shown together with an entire set of items within a given genre. Participants could switch between the two modes by pressing a key. (b) Top view of the experimental setup. Four participants in a single session unit were seated around a round table such that they could not see each other's responses. (c) An example bet distribution. Participants stacked the coins, which were visualized as a histogram on the betting axis. Participants were told that they would only win coins that had been bet on the actual SP (blue bar), which was unknown to them during the experiment. The ranking at the peak of the bet distribution was taken as a measure of SP estimation ($\hat{SP}$). (d) An example pair of bets in the initial and revision rounds. Left, although all four participants initially expressed their own SP estimate, they only saw their own SP estimate (gray histograms), but not the others' (empty histograms), during the initial round. Right, participants were exposed to none or a few (0–3) of the others' SP estimates and given a second chance to make their bet. In this example, two randomly chosen SP estimates of others (gray histograms on the second and third rows) were revealed. The original artwork images were replaced with hand-drawing images for an illustrative purpose in here.

On each initial-round trial, four participants within a given unit viewed an artwork for as long as they wanted; they indicated their best guess ($r_{best}$) of the item's ranking by placing the maximal bet at that rank and expressing the confidence in their guess by piling their bets around the best guess. The coins betted over ranks were displayed to participants as a histogram, which will henceforth be referred to as a "bet distribution" (as illustrated in Fig 4C). The sum of coins was always fixed at 100, and the bet distribution approximately followed a discretized normal distribution over 24 ranks, which means the coins that betted on the ranking (r: 1~24) is as follows:

$$\text{Betted coins at r} = C \frac{1}{\sqrt{2\pi\sigma^2}} e^{-\frac{(r-r_{best})^2}{2\sigma^2}}$$

where r is a rank ranging between 1 and 24, $r_{best}$ is the rank with the maximal bet, σ is the standard deviation of the distribution, and C is the normalization factor that makes the sum of coins 100. Participants manipulated the location and width of the bet distribution by adjusting $r_{best}$ using the "4" and "6" keys on the number pad, and σ using the "2" and "8" keys, respectively. Participants could adjust σ as one of the 19 values: 0 or 1 / (2.5 * (1−0.05 * k)), where integer k ranged from 1 to 18. When the level of σ was 0, all coins had been betted at $r_{best}$. We treated $r_{best}$ as an SP estimate, $\hat{SP}$. When adjusting the location and width of the bet distribution, participants were permitted to review the entire set of 24 artworks within a given genre (as shown in Fig 4B), which helped them to rank a given artwork by comparing it with all other artworks within the genre by clicking on a "review" button whenever needed. In the initial round, however, participants were neither allowed to see their partners' screens nor were they informed of their partners' bets. Each trial was completed when all four participants had finalized their bets, and the next trial then began, resulting in a mean trial duration of 14.3 s (standard deviation = 9.5 s).

The procedure for the revision-round trials was identical to that for the initial-round trials except that participants were, in most of the trials, informed of the coin distributions made by their partners in the preceding initial round. To investigate how participants' updating of their SP estimates depended on the number of others' SP estimates, we varied the number of bet distributions revealed to participants over trials by showing one, two, three, or no (zero) distributions at an equal proportion (25%). The no-distribution condition was included to characterize the changes of each individual's personal estimation that may have occurred due to repeated guesses made on the same artwork (but these data were not used in this study). We also controlled the frequency of exposure of any given partner not to be larger than those of other partners across trials (see S1 Table for details).

The presentation order of the 24 artworks was randomized separately and independently across participants in the same session. However, within each participant, the order of item presentation was identical between the initial round and the revision round. Once participants had completed the initial and revision rounds within a given genre, we informed them of their overall performance by showing them how many coins they had won so far. This helped them to remain motivated to perform the task as well as they could. The next session with a new genre then began. The presentation order of genres was randomized across session units.

## Definition of actual SP

To define the actual SP values of the individual items used in the main experiment, we conducted an online survey in which many (100) people provided their own personal preferences for each item. Those who participated in this survey did not participate in our main experiment and were paid 10,000 KRW for their participation in the survey. They indicated their

preference for a given item by choosing one of the integer scores ranging from 0 to 10. For normalization across participants and genres, they were asked to scale their preference scores such that the least preferred item to be 0 and the most preferred item to be 10 in each genre (24 items). We ranked the items based on the across-individual average of preference scores such that actual SP values of 1 (top rank) and 24 (bottom rank) were assigned to the items with the greatest and smallest average preference scores, respectively. This procedure was implemented separately for each of the five genres.

To confirm the validity of the scores acquired via the online survey, we also gathered personal preference data from a separate population of participants via an offline survey. A total of 225 people who visited a local museum participated in this offline survey upon request. They saw the 24 items of each genre simultaneously and marked the 5 most preferred items and the 5 least preferred items. For each item, we calculated the preference score by subtracting the number of people who disliked the item from the number of the people who liked it. Then, we took the ranking of the preference scores as the actual SP values. The rankings determined by the offline survey were highly correlated with those by the online survey (Spearman $r = 0.885$). More importantly, the main results regarding the utility of the accuracy cues were consistent between the online and offline surveys (see S3 Text for details).

### Regression model for social revision of SP estimates

To quantitatively capture the contributions of the accuracy cues to the modulation of accepting others' SP estimates, we built a basic regression model, as defined in Eq 1 in the Results section. For the sake of simplicity, Eq 1 lacks several details, including the subscripts that represent individuals ($i$) and items ($j$), and the dummy values that were assigned to the accuracy cues. The full description of the basic regression model is as follows:

$$\widehat{SP^r}_{ij} = \widehat{SP^m}_{ij} + W_{ij}\left(\widehat{SP^o}_{ij} - \widehat{SP^m}_{ij}\right) + B + \epsilon_{ij};$$

$$W_{ij} = W_0 + \beta_{cm}\, c^m_{ij} + \beta_n\, n^o_{ij} + \beta_a\, a^o_{ij} + \beta_{co}\, c^o_{ij}$$

where the dummy values of the accuracy cues were defined as follows:

$$c^m_{ij},\; c^o_{ij} \in \{-0.5,\; 0,\; 0.5\};\, a^o_{ij} \in \{-0.5,\; 0,\; 0.5\};\, n^o_{ij} \in \{-0.5,\; 0, 0.5\}$$

We determined the dummy values for the confidence accuracy cues ($c^m_{ij}$, $c^o_{ij}$) by assigning -0.5 to the raw measurements of confidence that were smaller than their median value, 0 to those that were equal to the median value, and +0.5 to those greater than the median value. Here, the median value was defined from the raw confidence measurements within each individual. These assignments allowed us to directly interpret the regression coefficients $\beta_{cm}$ (and $\beta_{co}$) as the difference in acceptance modulation between the trials in which individual (others') confidence was smaller than the median and those in which individual (others') confidence was greater than the median. The dummy values for the agreement accuracy cues ($a^o_{ij}$) were determined by a similar procedure to that used for the confidence accuracy cues, by assigning -0.5, 0, and +0.5 to the measurements smaller than, equal to, and greater than the median value, respectively. However, this procedure was different from the value assigning for confidence in two important aspects. First, the dummy values could be assigned only in trials in which the number of others' SP estimates was more than one. Second, the median value was not defined by using the raw agreement measurements within each individual, but by using raw agreement measurements across individuals within each of the two number of others conditions (n = 2 or n = 3) because the median of standard deviations (reciprocals of agreement) depends on the number of samples (3.18 for n = 2 and 4.16 for n = 3). Unlike the confidence

and agreement cues, the number cue has categorical values and thus did not require the median-split procedure. Instead, we assigned -0.5, 0, and 0.5 for n = 1, 2, and 3, respectively. These assignments allow $\beta_n$ to be interpreted as the change in the difference in acceptance modulation as the number of others' SP estimates increases by 2 steps.

## Permutation test for differences in mean between two groups

To evaluate the validity and utility of the accuracy cues, we tested whether the means of deviances significantly differed between two subgroups of a single population, split by the value of one of the accuracy cues. For example, we wanted to test whether the mean deviance in high-confidence trials is smaller than that of low-confidence trials (Fig 2D–2G). For statistical testing, we considered two things. First, we expected the difference in mean deviance between the two groups of trials to be in a particular direction, such that the mean deviance would be smaller in the subgroup with a high (level) accuracy cue (e.g. high-confidence trials) than in the subgroup with a low (level) accuracy cue (e.g. low-confidence trials). Thus, we used one-tailed p-value to assess statistical significance. Second, we expected that within-group data were unlikely to be normally distributed, which means that a two-sample t-test was not appropriate. Thus, we carried out a non-parametric one-tailed permutation test by approximating the sampling distribution of differences between means, and calculated the p-values from that distribution using the following steps. First, the subgroup labels were randomly re-assigned (permutated) for all trials. Second, the difference in mean deviance between the re-assigned subgroups was calculated, which resulted in a sample of the sampling distribution. Third, the first and second steps were repeated until a sufficiently large number of samples (10,000 times) was obtained. Finally, the p-value for the observed mean difference was calculated from the sampling distribution as follows:

$$p \; value = \frac{Number \; of \; samples \; in \; sampling \; distribution \; larger \; than \; observed \; difference \; between \; means \; + \; 1}{Total \; number \; of \; samples \; in \; sampling \; distribution + 1}$$

## Supporting information

**S1 Text. Details of the hierarchical regression model.**
(DOCX)

**S2 Text. Alternative procedures for replicating the main findings.**
(DOCX)

**S3 Text. Utility analysis based on two independent survey results.**
(DOCX)

**S4 Text. Weight on advice analysis.**
(DOCX)

**S1 Table. Conditions of exposing others' SP estimates.**
(DOCX)

## Acknowledgments

We acknowledge Juhyun Eune and Jungwon Ryu for contribution to design of the experiment and select the items used in the experiment.

## Author Contributions

**Conceptualization:** Jaeseob Lim, Sang-Hun Lee.

**Data curation:** Jaeseob Lim.

**Formal analysis:** Jaeseob Lim.

**Funding acquisition:** Sang-Hun Lee.

**Investigation:** Jaeseob Lim, Sang-Hun Lee.

**Methodology:** Jaeseob Lim, Sang-Hun Lee.

**Project administration:** Sang-Hun Lee.

**Supervision:** Sang-Hun Lee.

**Validation:** Jaeseob Lim, Sang-Hun Lee.

**Visualization:** Jaeseob Lim, Sang-Hun Lee.

**Writing – original draft:** Jaeseob Lim, Sang-Hun Lee.

**Writing – review & editing:** Jaeseob Lim, Sang-Hun Lee.

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
