## [Decision Letter · Decision Letter 0]

30 Jun 2020

PONE-D-20-12298

Utility and use of accuracy cues in social learning of crowd preferences

PLOS ONE

Dear Dr. Lee,

Thank you for submitting your manuscript to PLOS ONE. After careful consideration, we feel that it has merit but does not fully meet PLOS ONE’s publication criteria as it currently stands. Therefore, we invite you to submit a revised version of the manuscript that addresses the points raised during the review process.

We look forward to receiving your revised manuscript.

Kind regards,

Pablo Brañas-Garza, PhD Economics

Academic Editor

PLOS ONE

Journal Requirements:

2. We note that Figures 2 and 4 in your submission contain copyrighted images. All PLOS content is published under the Creative Commons Attribution License (CC BY 4.0), which means that the manuscript, images, and Supporting Information files will be freely available online, and any third party is permitted to access, download, copy, distribute, and use these materials in any way, even commercially, with proper attribution. For more information, see our copyright guidelines: http://journals.plos.org/plosone/s/licenses-and-copyright.

a)     You may seek permission from the original copyright holder of Figures 2 and 4 to publish the content specifically under the CC BY 4.0 license.

Reviewers' comments:

Reviewer's Responses to Questions

**Comments to the Author**

1. Is the manuscript technically sound, and do the data support the conclusions?

Reviewer #1: Yes

Reviewer #2: No

2. Has the statistical analysis been performed appropriately and rigorously? 

Reviewer #1: Yes

Reviewer #2: No

3. Have the authors made all data underlying the findings in their manuscript fully available?

Reviewer #1: Yes

Reviewer #2: Yes

4. Is the manuscript presented in an intelligible fashion and written in standard English?

Reviewer #1: Yes

Reviewer #2: Yes

5. Review Comments to the Author

Reviewer #1: This paper studies how the opinion of others affect of our rankings. The authors consider an experiment in which people are asked to judge paintwork according to their "social ranking" and then exposed to rankings/opinions by others, including confidence of others. The paper shows that people adapt their opinions when receiving information. The manuscript is well-written and the experiment is well-executed in my view.

Reviewer #2: The paper asks both how people should and actually do incorporate other people's beliefs into their beliefs of others. A contribution of the paper is to extend previous work on beliefs that are "factually correct" to beliefs that are "socially correct".

While I like the research question and theoretical analysis, I see a major flaw in the experimental design that renders most empirical conclusions uncertain. The empirical analysis hinges on how well the experimental subjects -- students -- predict how actual people ranked paintings. However, the experiment deceived students: actual people never ranked the paintings. So in the empirical analysis the paper replaces this actual ranking with students' predictions of it: unfortunately, we do not know whether these predictions are any good. Because the empirical analysis would need these actual rankings, but has only the predicted ones, the empirical conclusions are uncertain. Unless the authors can produce such an actual ranking and then use this data to redo the analysis, I cannot recommend publication.

If they can produce it, then the empirical analysis would also need revision. For instance, within a person, each ranking is not independent, and so this regression assumption is in doubt (at the minimum, the paper needs to cluster standard errors by student). Further, the data are truncated at 1 and 46 so for low and high ranking paintings the errors are not normally distributed.

6. PLOS authors have the option to publish the peer review history of their article (what does this mean?). If published, this will include your full peer review and any attached files.

Reviewer #1: No

Reviewer #2: No

---

## [Author Response · Author response to Decision Letter 0]

14 Sep 2020

We uploaded our point-to-point responses to the reviewers and editor's comments as a separate file, which was labeled as 'Response to Reviewers'. Please refer to that file.

---

## [Decision Letter · Decision Letter 1]

7 Oct 2020

Utility and use of accuracy cues in social learning of crowd preferences

PONE-D-20-12298R1

Dear Dr. Lee,

We’re pleased to inform you that your manuscript has been judged scientifically suitable for publication and will be formally accepted for publication once it meets all outstanding technical requirements.

Kind regards,

Pablo Brañas-Garza, PhD Economics

Academic Editor

PLOS ONE

Additional Editor Comments (optional):

Reviewers' comments:

Reviewer's Responses to Questions

**Comments to the Author**

1. If the authors have adequately addressed your comments raised in a previous round of review and you feel that this manuscript is now acceptable for publication, you may indicate that here to bypass the “Comments to the Author” section, enter your conflict of interest statement in the “Confidential to Editor” section, and submit your "Accept" recommendation.

Reviewer #1: All comments have been addressed

2. Is the manuscript technically sound, and do the data support the conclusions?

Reviewer #1: Yes

3. Has the statistical analysis been performed appropriately and rigorously? 

Reviewer #1: Yes

4. Have the authors made all data underlying the findings in their manuscript fully available?

Reviewer #1: Yes

5. Is the manuscript presented in an intelligible fashion and written in standard English?

Reviewer #1: Yes

6. Review Comments to the Author

Reviewer #1: I am satisfied with the revision as the authors have addressed my comments. They have also made substantial changes in the manuscript, in line with the suggestions of the other referee.

7. PLOS authors have the option to publish the peer review history of their article (what does this mean?). If published, this will include your full peer review and any attached files.

Reviewer #1: No

---

## [Editor Report · Acceptance letter]

12 Oct 2020

PONE-D-20-12298R1 

Utility and use of accuracy cues in social learning of crowd preferences 

Dear Dr. Lee:

I'm pleased to inform you that your manuscript has been deemed suitable for publication in PLOS ONE. Congratulations! Your manuscript is now with our production department. 

Kind regards, 

on behalf of

Dr Pablo Brañas-Garza 

Academic Editor

PLOS ONE